# MS-222 and Propofol Sedation during and after the Simulated Transport of Nile tilapia (*Oreochromis niloticus*)

**DOI:** 10.3390/biology10121309

**Published:** 2021-12-10

**Authors:** Luís Félix, Rita Correia, Rita Sequeira, Cristiana Ribeiro, Sandra Monteiro, Luís Antunes, José Silva, Carlos Venâncio, Ana Valentim

**Affiliations:** 1Instituto de Investigação e Inovação em Saúde (i3s), Universidade of Porto, 4200-135 Porto, Portugal; 2Laboratory Animal Science, Instituto de Biologia Molecular e Celular (IBMC), Universidade do Porto, 4200-135 Porto, Portugal; 3Centre for the Research and Technology of Agro-Environmental and Biological Sciences (CITAB), University of Trás-os-Montes and Alto Douro (UTAD), 5000-801 Vila Real, Portugal; smonteir@utad.pt (S.M.); lantunes@utad.pt (L.A.); cvenanci@utad.pt (C.V.); 4Institute for Innovation, Capacity Building and Sustainability of Agri-Food Production (Inov4Agro), University of Trás-os-Montes and Alto Douro (UTAD), 5000-801 Vila Real, Portugal; 5School of Agrarian and Veterinary Sciences (ECAV), University of Trás-os-Montes and Alto Douro (UTAD), 5000-801 Vila Real, Portugal; al65994@utad.eu (R.C.); jasilva@utad.pt (J.S.); 6School of Life and Environmental Sciences (ECVA), University of Trás-os-Montes and Alto Douro (UTAD), 5000-801 Vila Real, Portugal; al64339@utad.eu (R.S.); al64215@utad.eu (C.R.); 7Animal and Veterinary Research Centre (CECAV), University of Trás-os-Montes and Alto Douro (UTAD), 5000-801 Vila Real, Portugal

**Keywords:** propofol, MS-222, fish transportation, sedation, stress

## Abstract

**Simple Summary:**

Sedation has been shown to reduce stress and damage during live fish transportation. However, there is a lack of information regarding MS-222 alternatives. The present study aimed at assessing the effects of propofol during a 6-h simulated transportation of Nile tilapia (*Oreochromis niloticus*) focusing on physiology, histology, behavior, and meat integrity. Fluctuations in water chemistry, glucose, hematological parameters, and behavioral response were found in propofol-treated animals (although it was normalized after the recovery period). In general, the findings highlight the potential use of propofol as a beneficial mediator of transportation-induced stress in fish contributing to improve fish welfare.

**Abstract:**

The use of anesthetics has been suggested as a strategy to hamper live fish transport-induced stress. Still, there is insufficient data available on the use of alternative anesthetics to MS-222. This study investigated the use of propofol to mitigate stress in Nile tilapia (*Oreochromis niloticus*, 143.8 ± 20.9 g and 20.4 ± 0.9 cm) during a 6 h simulated transport. Individuals (n = 7) were divided into three groups: control, 40 mg L^−1^ MS-222, and 0.8 mg L^−1^ propofol. A naïve group non-transported was also considered. During the 6 h transport and 24 h after, the response to external stimuli, opercular movements, water quality parameters, behavior, blood hematology and other physiological values, the histopathology of the gills, the quality of the fillet, and oxidative-stress changes in gills, muscle, brain, and liver were evaluated. Propofol increased swimming activity of fish but decreased opercular movements and responses to external stimuli, indicating oscillations of the sedation depth. Water pH and glucose levels increased, while hematocrit (HCT) and lactate decreased in propofol groups at 6 h. At this time-point, MS-222 also induced a decrease in the HCT and lactate levels while increasing cortisol levels. Despite these effects, the stress-related behaviors lessened with anesthetics compared to the control group. After the recovery period, physiological responses normalized in animals from both anesthetic groups, but the control still had high cortisol levels. Overall, propofol is a good alternative for the transportation of this species, showing efficient sedation without compromising health or fillet quality. However, further pharmacodynamics and pharmacokinetics knowledge is required to support its use in aquaculture settings.

## 1. Introduction

In recent years, there has been a growing concern related to fish welfare in research and aquaculture [1]. Consumers have become more demanding regarding the welfare of production animals and, although slowly, fish are no exception [2,3]. The pressure of public opinion, scientists, policy makers, and other stakeholders resulted in specific directives and recommendations to regulate the use of fish for a variety of purposes (mainly research and aquaculture). Indeed, there is various evidence that this vertebrate taxon can experience pain [4,5]. In general, handling, transport, and husbandry procedures are the key stressful events that can have negative impacts on fish welfare [1]. The transport of live fish, for instance, is a crucial process for various research and aquaculture purposes [6,7]. Many studies have emphasized the physiological stress responses induced by this practice associated with changes in water quality [6]. The stress response is characterized by the stimulation of the hypothalamic-pituitary-interrenal axis with the release of corticosteroids and catecholamines [8,9,10,11]. This induces the activation of a cascade of events that may culminate in deleterious effects on fish health, increasing disease susceptibility and mortality [8,12] and thereby reducing the profitability and welfare of fish. In this view, it is essential to refine the appropriate transport protocol, inducing the least stress possible.

One of the mitigation strategies that has been implemented to reduce transport-induced stress is the use of anesthesia [13,14]. Sedation of fish can lower the fish’s metabolism resulting in improved water quality and can lower stress levels related to transport procedures such as handling, turbulence, and aggressive social interactions [14,15] while also maintaining meat quality [16]. Among the broad spectrum of fish anesthetics, the most used is tricaine methanesulfonate (MS-222) [13,14], although species differences in pharmacokinetics and pharmacodynamics have been described [17]. In addition, a report has shown this compound to be aversive to some fish species [18], and further studies demonstrated that MS-222 might elicit immune-biochemical responses (increased cortisol, altered oxidative status and immune system) in different fish species under transport conditions [19,20,21]. In general, MS-222 may compromise the health and welfare of the fish, requiring further studies to better understand how this and other anesthetics affect welfare during fish transport.

Anesthetics such as propofol (2,6-diisopropylphenol), a widely used GABA_A_ agonist for human and veterinary anesthesia, have been evaluated experimentally in different fish species [22,23,24,25,26]. It is a short-acting, rapid recovery, and fast metabolized anesthetic agent with a lack of cumulative effects reported in Nile tilapia (*Oreochromis niloticus*) [24], the second most farmed fish worldwide [27]. In addition, although recent studies have characterized propofol induction and recovery in this species [28,29], no thorough research has been conducted so far using propofol for transport conditions of Nile tilapia, while it has been validated for the transport of other species [26]. Still, due to between- and within-species drug responses differences, anesthetics must be investigated on a species-by-species basis to assure its safe use [30]. Therefore, this study aimed to evaluate the efficacy of propofol during and after the simulated transportation of Nile tilapia in comparison to MS-222, considering alterations in physiology, histology, behavior, and meat integrity. Overall, these will provide information regarding fish welfare/health and, consequently, productivity and fillet quality following transportation of tilapia under these compounds.

## 2. Materials and Methods

### 2.1. Reagents and Solutions

Tricaine methanesulfonate (CAS 886-86-2) was purchased from Sigma-Aldrich (Sintra, Portugal) and was prepared as a stock solution at a concentration of 4000 mg L^−1^ (~15.30 mM) in deionized water and buffered with sodium bicarbonate to a pH of 7.4. This solution was kept in an amber bottle and refrigerated until used. Propofol (Propofol-Lipuro 1%, 10 mg mL^−1^) was purchased from B. Braun (Barcarena, Portugal). Anesthetic solutions were freshly prepared prior to transport simulation by diluting stock solutions to the desired nominal concentration in the system water. Unless stated, all other chemicals were of the highest grade available and obtained from Sigma-Aldrich (Sintra, Portugal).

### 2.2. Experimental Animals

This study was carried out at the University of Trás-os-Montes and Alto Douro (UTAD), Vila Real, Portugal. All the procedures were approved by National authority DGAV (Direção Geral de Alimentação e Veterinária) through the project license 014703/2017-06-16 and in agreement with the European Directive on the protection of animals used for scientific purposes (2010/63/EU) and its transposition to the Portuguese law (Decreto-lei 113/2013) ensuring minimal animal stress and discomfort. A total of 55 Nile tilapia (*Oreochromis niloticus*) (body weight of 143.8 ± 20.9 g and length of 20.4 ± 0.9 cm measured after euthanasia) raised in UTAD Experimental Research Station (Vila Real, Portugal), were acclimated for one week at 25 ± 1 °C in two 500-L recirculating tanks filled with dechlorinated Vila Real tap water at a density of 1 fish/15 L with continuous aeration, biological and mechanical filtration, and maintained under a 14-h light, 10-h dark cycle. During the acclimation period, fish were fed daily with a control diet (35.8% of protein and 8.3% fat) [31], corresponding to around 1.5% of their weight; 24 h prior to the initiation of the experiments, fish were starved. The renewal of 50% of the water from the system and feces removal was performed every five days to minimize the build-up of nitrates and nitrites and to maintain water quality.

### 2.3. Simulated Transport

For the transport simulation experiment (Figure 1), individual fish were randomly and rapidly net transferred to plastic boxes of 10 L containing 8 L of system water with continuous aeration (7.78 ± 0.11 mg O_2_ L^−1^) and distributed into three different experimental groups: control (no sedation), MS-222 (sedated with 40 mg L^−1^/0.15 mM) and propofol (sedated with 0.8 mg L^−1^/0.0045 mM). The tested concentration of MS-222 was related to the sedation induction previously observed in this species [32], while the concentration used for propofol was selected based on pilot experiments (data not shown) and on a previously published work [24]. In all cases, the selected concentrations of the anesthetics induced light sedation characterized by disorientation, reduced activity, reactivity, and ventilation rate while maintaining the normal equilibrium [33,34]. A naïve group was also included which did not receive any treatment during the experiment being maintained in the 500-L tanks. To simulate transport conditions, the plastic boxes were placed over a table vibrating at a mean 0.62 ± 0.24 m s^−2^ and producing a noise of 77.8 ± 0.47 dB, thus mimicking the conditions observed in the transport of live animals [35,36,37]. In addition, Nile tilapia were transport simulated for 6 h to mimic the common duration of short and regional transportation (range between 4 and 8 h, reviewed in [6]). At the end of the simulated transport protocol, seven animals from each treatment groups (n = 7) and six from the naïve group (n = 6) were sacrificed by stunning them with a blow to the head followed by rapid decapitation. The remaining fish subjected to transport (n = 7) were net transferred to a 200-L clean-water tank and allowed to recover for 24 h and then euthanized and sampled as well. Seven animals from the naïve group were also sampled at this time. No food was supplied during the recovery period, and this was established based on the complete recovery of physiological functions (no changes in plasma cortisol concentrations) following acute stress in this species [38].

### 2.4. Clinical and Sedation Evaluation

During the simulated transport conditions described above, the fish from the different treatments were monitored for equilibrium and respiratory activity with observations at 1 min (assumed hereafter as 0 min), 0.5, 1, 3, and 6 h (n = 7). The equilibrium was evaluated as absent or present (0 or 1) while the respiration (opercular movement rate) was assessed by counting the opercular movements for 60 s. In addition, reactions to visual and vibrational cues (moving the net on the opposite side of the tank) and to tactile stimulus were assessed at these time-points to evaluate the level of consciousness and to confirm sedation. The tactile stimulus was characterized by gently touching the dorsal side of the animal antero-posteriorly with a plastic tube; if the animal did not respond, a pinch in the caudal peduncle was gently applied with forceps. All of these responses were evaluated at 0.5, 1, 3, and 6 h by the same person.

### 2.5. Water Quality Assessment

Water samples were collected before the simulated transportation period (baseline values) and at 3 and 6 h of transport simulation to monitor some chemical parameters of the system water. The pH, temperature, and dissolved oxygen were assessed using a multi-parameter water quality analyzer (HQ40d, Hach, Loveland, CO, USA).

### 2.6. Behavioral Assessments

During the simulated transportation (at 0, 0.5, 1, 3 and 6 h), at the end of this transport (6.5 h) and after 24 h of recovery, fish (n = 7) were individually video recorded with a Canon Full-HD (1920 × 1080 pixels) camcorder (Canon Legria HF R606) placed in front of the plastic box and mounted on tripods which were positioned at 1 m from the tanks to avoid interferences on the test fish. Each fish was recorded for behavioral responses for 5 min at each time-point. From these video recording, behavioral data were measured by the duration the fish engaged in specific behavior and/or by the number of times (frequency) the specific behavior was observed, according to the implemented ethogram (Table 1), based on the most common types of behaviors observed in this species [39]. A single-blinded observer for the treatments used the open-source event-logging software Behavioral Observation Research Interactive Software (BORIS v7.6 [40] to analyze each fish.

### 2.7. Blood Analyses

By the end of the transportation period (at 6 h) and following the recovery period (24 h post-transport end), blood samples were taken from each animal immediately after euthanasia through caudal vein puncture using heparinized syringes. The blood was immediately transferred up to 2/3 of a microhematocrit capillary tube and centrifuged (Heraeus Pico 17 Centrifuge, Thermo Scientific, Waltham, MA, USA) at 10,000× *g* for 5 min and the percentage of hematocrit (HCT) was calculated. An additional portion (10 μL) was used to determine hemoglobin (Hb) concentrations using the Drabkin’s reagent (0.6 mM potassium ferricyanide, 0.77 mM potassium cyanide, 1 M potassium dihydrogen phosphate, 0.1% Triton X-100 and pH 7–7.4) [41]. Following dark incubation at 4 °C for 1 h, samples were left for 15 min at room temperature (the method is stable for up to 5 h [41]) followed by centrifugation at 10,000× *g* and read at 540 nm. The absorbance of the solution was further converted to Hb concentration according to the equation previously described [41]. The remaining blood was collected to 2.0 mL microtubes, allowed to clot, and centrifuged at 5000× *g* for 10 min (4 °C, Sigma model 3K30, Osterode, Germany) to collect the plasma which was stored at −20 °C for later analysis. Cortisol was double extracted with five volumes of diethyl ether from 100 μL of plasma and agitated for 1 min. Samples were then centrifuged (2000× *g*, 5 min) and the organic layer was collected into new tubes. The organic phase was further evaporated under a fume hood and the dried organic was reconstituted in 500 μL of phosphate-buffered saline (PBS; pH 7.4). For the analysis of cortisol levels, 50 μL of the reconstituted samples were used and assayed at 405 nm by following the ELISA kit instructions (product 500360; Cayman Chemical Company, Ann Arbor, MI, USA). Serum glucose and lactate concentrations were determined by using an enzymatic colorimetric test (SpinReact kit nr 1001200 or nr 1001330, respectively, SpinReact, Girona, Spain) following the manufacturer’s instructions with slight modifications. Serum samples (10 μL) were mixed with 200 μL of working reagent and incubated for 5 min at 37 °C before being read at 340 nm against a standard curve of glucose (0–10.0 mg mL^−1^) or at 505 nm against a standard curve of lactate (0–1.0 mg mL^−1^). Spectrophotometric measurements were made on a microplate reader (PowerWave XS2, Bio-Tek Instruments, Winooski, VT, USA).

### 2.8. Fillet Quality and Glucose Based-Glycogen Content

The pH, color, redox potential (Eh), and drip loss were evaluated in samples collected at the end of the transportation period and following the recovery period to assess changes in the fillet quality. For the redox potential determination, a portable digital potentiometer (WTW 330i) with a SenTix ORP platinum electrode was used. Measurements were taken directly in the epaxial body muscle after post-mortem (15 min) and the readings were taken after 3 min for the stabilization of values. The pH was also measured directly in the epaxial body muscle (fillet) with a portable digital potentiometer (Hanna HI 98163) post-mortem (15 min). The color measurements were taken at the same time-point with a handheld CR-10 colorimeter (Konica Minolta Sensing Inc., Osaka, Japan) used at 90° in the epaxial body muscle to record six different readings per sample to obtain the lightness (L*), redness (a*), and yellowness (b*) according to the CIELAB system. The a* and b* values were used to calculate hue (H_ab_ = 180 + tan^−1^ (b*/a*) and chroma (C_ab_ = (a*2 + b*2)^1/2^ representing the angular measurement and the expression of intensity and clarity of the color, respectively. The difference of color between treatments and baseline was calculated as ∆E = (∆L*2 + ∆a*2 + ∆b*2)^1/2^ where, in this case, ∆ refers to the differences between the naïve and the transport-based groups [42,43]. For the drip loss, fillet samples from the epaxial body muscle were weighed (Wi), vacuum packaged and stored at −20 °C for seven days. Afterward, wiped samples were weighed (Wf), the drip loss (%) was calculated as [(Wi − Wf)/Wi] × 100. The glucose based-glycogen content was determined, with slight modifications, as previously described [44]. Briefly, the frozen samples (130 mg) were solubilised with 1 mL 6.0 N KOH in boiling water for 5 min and precipitation was achieved by the addition of 1 mL 95% ethanol and 100 μL 10% K_2_SO_4_-saturated solution. Following a centrifugation step (3000 rpm for 5 min), the glucose released was estimated at 540 nm on a microplate reader (PowerWave XS2, Bio-Tek Instruments, Winooski, VT, USA) by the 5% phenol-sulfuric acid method (1:5 *v*:*v*) [45].

### 2.9. Histological Analysis of the Gills

The gills are responsible for gas exchange and osmoregulation. It is in direct contact with the anesthetic, constituting a putative target organ for its side effects. To evaluate this, one gill arch from each side of the body was randomly collected from each fish at 6 h after the beginning of the experiment and at 24 h after transport ended to assess its integrity by histopathological analysis. Briefly, after euthanasia, gill arches were fixed for 24 h in buffered formaldehyde as previously described [46]. Samples were then dehydrated with ascending grades of alcohol, cleared in xylene, and embedded in paraffin wax in a modular tissue embedding center (Leica EG 1160). Sagittal sections (3 μm thick) were obtained in a rotary microtome (Leica RM 2135), mounted on glass slides, and stained with hematoxylin and eosin (HE). Slides were further washed in running tap water, dehydrated in alcohol, cleared in xylene, and mounted with Entellan^®^ (Merck, Germany). Sections were observed using a Nikon (Tokyo, Japan) 600N light microscope. A qualitative evaluation was first performed, the histopathological changes grouped in two observed categorical patterns: circulatory (vasodilatation and oedema) and regressive (epithelial lifting, filament epithelium proliferation, and lamellar fusion) [47], and the prevalence index was determined. The magnitude of the histopathological changes was further assessed using a four degrees semi-quantitative severity gradation scale [48], considering the extent and severity of each lesion (i.e., the percentage of filaments with that type of lesion in each fish sampled). In general, the degrees attributed corresponded to: 0—no histopathological changes, 1—minimal, 2—mild, and 3—moderate pathological alterations. The histological alteration index (HAI) was further calculated from the sum of the degrees assigned.

### 2.10. Oxidative-Stress-Related Responses

Biochemical analyses were carried out by spectrophotometric (PowerWave XS2, Bio-Tek Instruments, Winooski, VT, USA) or spectrofluorometric analysis (Cary Eclypse, Varian, Palo Alto, CA, USA) at the end of the simulated transport and 24 h after as previously described [49,50]. Briefly, gill, muscle, brain and liver from each specimen were homogenized in 500 µL cold buffer (0.32 mM of sucrose, 20 mM of HEPES, 1 mM of MgCl_2_, and 0.5 mM of phenylmethyl sulfonylfluoride (PMSF), pH 7.4) [51] in a Tissuelyser II (30 Hz for 30 s—Qiagen, Hilden, Germany). Supernatants were collected after centrifugation at 12,000× *g* for 10 min at 4 °C in a refrigerated centrifuge (Sigma 3K30, Osterode, Germany) and protein was determined by the Bradford method at 595 nm with bovine serum albumin (BSA) as a standard [52]. The overall reactive oxygen species (ROS) generation was measured using the 2′,7′-dichlorofluorescein diacetate (DCFH-DA) probe at 485 nm (excitation) and 530 nm (emission) [51]. Superoxide dismutase (Cu/Zn-SOD) activity was assayed by measuring the inhibition of nitroblue tetrazolium (NBT) reduction at 560 nm [53]. Catalase (CAT) activity was followed by the decrease in the H_2_O_2_ absorbance at 240 nm [54]. The activity of glutathione peroxidase (GPx) and glutathione reductase (GR) was measured by following the oxidation and reduction, respectively, of NADPH at 340 nm [55]. The enzymatic activity of glutathione-s-transferase (GST) was measured at 340 nm using CDNB as substrate [56]. The reduced (GSH) and oxidised glutathione (GSSG) were derivatized with ortho-phthalaldehyde (OPA) and measured at 320 nm and 420 nm for excitation and emission wavelengths, respectively [57]. The GSH:GSSG ratio was used to describe the redox ratio (oxidative stress index, OSI). The content of malondialdehyde (MDA), the major product of lipid peroxidation, was estimated by the quantification of the MDA-TBA adducts at 530 nm with a correction for non-specific adducts at 600 nm [58].

### 2.11. Statistical Analysis

Sample size calculation was performed in G*Power 3.1 (University of Düsseldorf, Düsseldorf, Germany), assuming a two-tail analysis, type II error probability of α = 0.05, a power of 0.85, and an effect size of 0.8. As naïve group was sampled at the same time of the day, although at different days, data were merged into a single dataset for further analysis. The statistical analysis was performed using GraphPad Prism 8 (GraphPad Software, La Jolla, CA, USA). Normal distribution of the data and homogeneity of variances were assessed by the Shapiro–Wilk test and Brown–Forsythe test, respectively. In general, two-way or repeated measures (RM) analysis of variance (ANOVA) was employed to identify differences between groups and effects over time. The Greenhouse-Geisser correction against unequal variances was applied for RM ANOVA analysis and reported degrees of freedom were rounded to the nearest whole number. In the event of a significant interaction, the simple main effects were determined by Tukey’s (post hoc) pairwise comparisons. When data were not normally distributed, the Kruskal-Wallis H test followed by Dunn’s multiple comparison test was used for differences between groups, while differences over time were tested using the non-parametric Friedman test, the Wilcoxon matched-pairs signed-ranks test or the Mann-Whitney U test. Also, the Chi-square test was used to analyze binary data, followed by the Bonferroni type pairwise comparison as post-hoc. Outliers were detected using the ROUT outlier test with an average false discovery rate of 1% [59] and removed from the analysis.

## 3. Results

### 3.1. Propofol Reduced the Responses to External Stimuli

Overall, no mortalities were observed during or after the simulated transportation procedure, independently of the treatment. The temporal changes in the respiratory rate of fish from the different tested groups are shown in Figure 2. The RM ANOVA revealed that there was an effect of time (F_(2.56,15.36)_ = 39.54, *p* < 0.0001), groups (F_(2.32,13.91)_ = 24.12, *p* < 0.0001) and there is an interaction between the two factors (F_(3.80,21.23)_ = 8.775, *p* < 0.0001), thus simple effects of factors were analyzed. Thus, at 0 h, all the groups had a higher respiratory frequency in relation to the naïve group (*p* < 0.001). The fish exposed to MS-222 showed a 16% decrease in the respiratory rate in comparison to the control fish (*p* = 0.005) but this was not different from the propofol-exposed individuals. At 0.5 h, the respiratory rate of the animals exposed to MS-222 and propofol decreased to values similar to the naïve fish. However, a 34% increased rate was observed in the control fish compared to the naïve group (*p* = 0.008). Also, the propofol-treated animals had significantly lower opercular movements compared to the control (*p* = 0.005). After 1 h and 3 h of the transportation, the control fish remained with higher opercular movements in relation to the naïve (*p* = 0.002; *p* < 0.001), MS-222 (*p* = 0.021; *p* < 0.001) and propofol (*p* < 0.001) treated animals. By the end of the transportation period, the respiratory rate of the propofol-exposed fish was lower than that of the control (*p* = 0.003). Regarding time, control, MS-222 and propofol-exposed fish significantly decreased respiratory rate from 0 h to the other time-points (*p* < 0.001, *p* = 0.040 and *p* = 0.007, respectively). In addition, as shown in Table 2, the response to the visual stimulus at 0.5 h revealed a significant difference among the three tested groups (X^2^(3) = 8.143, *p* = 0.017) with propofol-treated animals showing a decrease in the response to this stimulus in comparison to the control fish (*p* = 0.005). The Bonferroni post hoc comparisons showed that these differences were maintained during the simulated transportation without statistical differences between the MS-222 and the control animals. When touch stimulation was analyzed, statistical differences (X^2^(3) = 14.13, *p* = 0.0009) were detected at 0.5 h with the propofol-treated animals showing no response to the touch on the lateral side (*p* = 0.0002) in comparison to the control and the MS-222 fish. These differences between groups were maintained in the following analyzed time-points. Regarding the caudal pinch, which was only applied when no response to the touch stimulus was observed, more than 50% of the propofol sedated animals responded in all time-points, while all control and MS-222 animals tested reacted to this stimulus (*p* < 0.05).

### 3.2. Water pH Changed over Time

The chemical characteristics of the water were evaluated during the transport procedure and the results are shown in Table 2. The RM ANOVA of the pH results revealed that there was no significant effect of time (F_(1.04,6.29)_ = 5.119, *p* = 0.062) nor between groups (F_(1.09,6.52)_ = 5.211, *p* = 0.057), but there was a significant interaction between both factors (F_(1.36,8.13)_ = 6.863, *p* = 0.024). Thus, the simple effects were analyzed to avoid the influence of each factor. Regarding time effect within each exposed group, the water pH of the control fish significantly increased at 6 h compared to the levels at 3 h (*p* = 0.023) while at 6 h the levels of water pH of the propofol-treated fish increased in relation to the levels of the beginning of the experiment (*p* = 0.023) and at 3 h (*p* = 0.010). Regarding treatment effect within each time, the MS-222-sedated animals had lower water pH levels at 3 h (*p* = 0.018) and 6 h (*p* = 0.028) compared to the control. Relative to the dissolved oxygen during the simulated transport, changes over time were observed (F_(1.14,6.81)_ = 15.91, *p* = 0.005) while no effects were observed regarding the treatments (F_(1.57,9.42)_ = 1.580, *p* = 0.252) and their interaction (F_(2.13,12.78)_ = 3.241, *p* = 0.070). In this context, an increase of about 6% was observed in the control group between 3 and 6 h (*p* = 0.029) while dissolved oxygen levels remained similar for MS-222 group. However, a decrease of about 8% was observed in the propofol-treated fish at 3 h in relation to the beginning of the transport (*p* = 0.024) while at 6 h the levels were similar to 0 h and different from the previous time-point (*p* = 0.041).

### 3.3. Propofol Induced an Increase in the Swimming Activity during Transport

During the simulated transport, different categorized behaviors (Table 1) were observed, and the results are shown in Table 3. Overall, swimming was affected at 0.5, 1 and 6 h (X^2^(2) = 10.60, *p* = 0.002, X^2^(2) = 7.117, *p* = 0.022 and X^2^(2) = 11.94, *p* = 0.0006, respectively) in the MS-222 and propofol-treated fish, which swam less often in comparison to the control fish (*p* < 0.05); no time effects were observed. Notwithstanding, propofol-treated animals showed a higher duration of this behavior at 0.5, 1, 3 and 6 h (X^2^(2) = 11.52, *p* = 0.0008, X^2^(2) = 11.45, *p* = 0.0009, X^2^(2) = 8.00, *p* = 0.013, and X^2^(2) = 13.07, *p* = 0.0002, respectively) in comparison to the MS-222 group. Relative to the effects over time, at the end of the exposure period, propofol-treated animals showed a significant increase in the duration of this behavior (X_F_^2^(5) = 12.34, *p* = 0.015) in comparison to the beginning of the experiment (*p* = 0.024). Post hoc tests for the incidence of bottom swimming behavior using the Bonferroni correction showed statistical differences between groups at 0.5 and 6 h (X^2^(2) = 8.00, *p* = 0.013, and X^2^(2) = 13.07, *p* = 0.0002, respectively) (*p* < 0.05). Propofol-treated animals at 0.5 h (*p* = 0.017) and 6 h (*p* = 0.026) and MS-222-exposed animals at 6 h (*p* = 0.020) showed a less incidence of this behavior in comparison to the control animals. Also, differences throughout time were detected for the control and MS-222 groups (X_F_^2^(5) = 11.50, *p* = 0.022 and X_F_^2^(5) = 10.66, *p* = 0.031, respectively). Although no significant changes were observed between groups for the duration of this action (*p* > 0.05), MS-222 treated fish (X_F_^2^(5) = 14.43, *p* = 0.006) swum less in the bottom at 6 h than at 0.5 h (*p* = 0.024). Relative to the inactive frequency, at 0 h no significant changes were observed. Yet, at the following time points, significant changes were observed between groups (X^2^(2) = 8.304, *p* = 0.010, X^2^(2) = 6.781, *p* = 0.027, X^2^(2) = 13.47, *p* < 0.0001, and X^2^(2) = 13.81, *p* < 0.0001, respectively for 0.5, 1, 3 and 6 h) with propofol showing less manifestation of this behavior in comparison to the control group (*p* < 0.041). At the end of the exposure period, MS-222-sedated animals also showed a decreased occurrence of this behavior compared to the control group (*p* = 0.043). MS-222 fish also showed less time inactive at 6 h than at 0.5 h (*p* < 0.05). Regarding the duration of the inactivity state, animals under propofol showed fewer periods of inactivity in comparison to the remaining groups at all time points (*p* < 0.05). Additionally, propofol animals spent less time inactive at 6 h than at 0 h (X_F_^2^(5) = 17.94, *p* = 0.001). Control animals did more erratic movements than MS-222 (*p* = 0.0271) and propofol (*p* = 0.038) fish at 1 h. Propofol animals at 0 h (*p* = 0.015) and 1 h (*p* = 0.032), and MS-222 animals at 1 h (*p* = 0.028) spent less time doing erratic movements than the control group (X^2^(2) = 7.844, *p* = 0.014 for 0 h, and X^2^(2) = 8.889, *p* = 0.007 for 1 h). Almost no erratic movements were detected for the MS-222 and propofol sedated individuals at 1 h, while the control animals decreased their reaction over time (X_F_^2^(5) = 13.54, *p* = 0.009) to almost no erratic movements after 3 h of the transportation process (*p* < 0.041). The analysis demonstrated that no changes were observed for the air stone breathing behavior while the incidence of turning behavior was affected by the treatments at 0.5 (X^2^(2) = 6.977, *p* = 0.024) and 1 h (X^2^(2) = 9.753, *p* = 0.004) with propofol-treated animals showing less incidence at both time-points in comparison to the control animals (*p* = 0.025 and *p* = 0.017, respectively). At 1 h, MS-222 also decreased the frequency of turns relative to the control fish (*p* = 0.025). MS-222 treated animals also showed a decrease in this behavior over time (X_F_^2^(5) = 16.16, *p* = 0.003) with significant changes observed between the first and last time-point of analysis (*p* = 0.005). Rubbing behavior was not different between groups, but it was decreased (X_F_^2^(5) = 11.48, *p* = 0.022) at 3 h relatively to the starting point in the control group (*p* = 0.041). Relative to the recovery period (Appendix A) there were no differences in none of the parameters studied, except at 6.5 h (X^2^(2) = 6.738, *p* = 0.037), when propofol-treated fish interacted more often with the objects (tubes, resistance, air stone, etc.) inside the tank than the control fish (*p* = 0.036).

### 3.4. General Stress Indicators Were Affected by Both MS-222 and Propofol

The changes in HCT (A) as well as the response of cortisol (B), glucose (C), and lactate (D) to the simulated transport are shown in Figure 3. The two-way ANOVA revealed an interaction between time and groups (F_(3,25)_ = 8.630, *p* = 0.0004) and an effect of time (F_(1,25)_ = 15.99, *p* = 0.0005) but no effect of group (F_(3,25)_ = 1.821, *p* = 0.169) on the HCT, and thus simple effects were analyzed. Regarding time, the HCT increased 27 and 47% for MS-222 and propofol, respectively, between the end of the transportation (6 h) and the recovery period (24 h) (*p* = 0.014 and *p* < 0.0001). At 6 h, the control fish had higher HTC compared to MS-222 (*p* = 0.013) and propofol animals (*p* = 0.009). In addition, the propofol sedated fish also presented lower levels than the naïve animals (*p* = 0.034). No differences between groups were observed at 24 h post-transport. Despite these differences in the HCT, no significant changes were observed for the hemoglobin levels between groups (Appendix A). Regarding cortisol levels, no significant effects were observed over time (F_(1,42)_ = 1.386, *p* = 0.246). However, an interaction between time and groups (F_(3,42)_ = 2.983, *p* = 0.042) and a significant effect was detected between groups (F_(3,42)_ = 7.184, *p* = 0.0005) with MS-222 showing higher mean levels of cortisol in comparison to the naïve (*p* = 0.010) and propofol (*p* = 0.048) groups at 6 h. At 24 h, only the control group showed increased cortisol levels compared to the naïve group (*p* = 0.012). Regarding the serum glucose levels, there was an interaction between the treatment and time (F_(3,29)_ = 8.292, *p* = 0.0004), and an effect of time (F_(1,29)_ = 22.94, *p* < 0.0001) and treatment (F_(3,30)_ = 6.369, *p* = 0.002); thus simple effects were analyzed. In this context, both control and propofol animals showed increased mean glucose in comparison to the naïve (*p* = 0.023 and *p* = 0.005, respectively), and the propofol fish had higher values than the MS-222 fish (*p* = 0.024) at 6 h. Also, control and propofol animals had higher levels of glucose at 6 h than after recovery at 24 h (*p* = 0.0003 and *p* = 0.026, respectively), where all groups had similar levels. Related to the serum lactate levels, an interaction between time and groups (F_(3,28)_ = 5.865, *p* = 0.004), an effect of time (F_(1,28)_ = 14.58, *p* = 0.0007) as well as an effect of group (F_(3,30)_ = 4.024, *p* = 0.016) was observed, with both MS-222 and propofol showing reduced lactate levels relative to the naïve group (*p* = 0.002 and *p* = 0.025, respectively) at 6 h. The values were similar between groups 24 h post-transport. In addition, both MS-222 and propofol-treated animals increased their lactate levels from 6 to 24 h (*p* < 0.007).

### 3.5. The Quality of the Fillet Was Not Affected by the Treatments

At the end of the transportation period and 24 h after, the quality of the fish fillet was assessed, and the results are shown in Appendix A. The pH was maintained similar among groups at all the timepoints evaluated. Still, statistical differences were found between time points (F_(1.36,25.17)_ = 92.4, *p* < 0.0001), as pH fillet decreased in the MS-222 and propofol sedated animals from 6 to 24 h. No significant changes were determined among groups nor between time-points for the redox potential, the color of the muscle fillet, the drip loss, and for the glucose-based glycogen content.

### 3.6. Gill Functionality Was Not Affected by the Anaesthetics’ Exposure

The microscopical evaluation of the gills, at 6 and 24 h, showed the presence of some histopathological changes (Figure 4A) in all fish, although with varying prevalence. The circulatory disturbances (vasodilatation (Vas) and oedema (OE)) were the most remarkable changes observed being recorded in over 50% of fish, at both 6 and 24 h (Figure 4B), without significant changes between groups (*p* > 0.05). Epithelial lifting (EL), filament epithelium proliferation (FEP) and lamellar fusion (LF) were regularly registered in around 15% of the evaluated specimens. Based on these observations, the HAI was calculated, and the results are shown in Figure 4C. Overall, the control group presented the highest average HAI value in comparison to the naïve group, although not significantly different and below the point at which the functionality of the organ would be compromised (HAI < 10).

### 3.7. Propofol Reduced ROS and GST Activity in Gills and GR Activity in Liver

At the end of the simulated transport and after the recovery period, different oxidative stress-related parameters were evaluated in the gills, muscle, brain, and liver. The significant differences between treatments at each specific time-point are presented in Figure 5 and all the results are shown in Appendix A. Propofol-treated animals showed reduced levels of ROS (Figure 5A) in the gills in comparison to the control group (*p* = 0.008) at 6 h (X^2^(3) = 10.37, *p* = 0.016), while no differences were observed after the recovery period nor over time. The GST activity (Figure 5B) was higher in the control group relative to the naïve (*p* = 0.010) and propofol-treated animals (*p* = 0.026) at 6 h (X^2^(3) = 11.76, *p* = 0.008) while no changes were detected at the end of the recovery period nor between both collection time-points. Regarding the glutathione levels in the muscle, GSH levels were similar among groups, but significantly higher levels (F_(1,45)_ = 6.939, *p* = 0.012) were observed in the control fish at 6 h compared to the ones at 24 h (*p* = 0.023). Regarding the OSI, a significant effect of time (F_(1,11)_ = 5.860, *p* = 0.034) and an interaction between time and group was detected (F_(3,11)_ = 6.021, *p* = 0.011), but no effect of group was found. The OSI decreased from 6 h to 24 h in the control group (*p* = 0.007). Regarding oxidative evaluation in the brain, there were no significant differences for the GR activity between time and groups but a significant interaction was determined between factors (F_(3,13)_ = 4.577, *p* = 0.021). GR activity increased at the end of the recovery period in comparison to the 6 h period in the naïve group (*p* = 0.005, Appendix A). The GSSG levels (Figure 5C) were also affected at 6 h (F_(3,23)_ = 5.766, *p* = 0.004) with control animals showing higher activity in relation to the naïve (*p* = 0.007) and propofol-treated (*p* = 0.006) animals. Regarding time effect (F_(3,13)_ = 4.868, *p* = 0.015), propofol animals had higher GSSG levels after the recovery period when compared to the 6 h values (*p* = 0.025) (Appendix A). Despite these differences, the OSI was similar. As for the liver analysis, there was an effect of time (F_(1,34)_ = 4.594, *p* = 0.039) on the catalase activity with a decrease observed between 6 h and 24 h for the naïve fish (*p* = 0.037, Appendix A). In addition, there was an effect in the glutathione reductase activity at 6 h (X^2^(3) = 8.596, *p* = 0.035, Figure 5D) with the control group showing higher activity in comparison to the naïve group (*p* = 0.035), albeit these differences were abolished by 24 h.

## 4. Discussion

The transport refinement of live fish is a crucial process for the survivability, animal welfare, and profitability of fish in aquaculture. Many studies emphasize the physiological stress responses and their consequences induced by the common transport procedures. The use of anesthetics is a relatively new research subject in live-fish transport, with the use of MS-222 potentially compromising the health of fish species such as crucian carp (*Carassius auratus*) [19], gilthead sea bream (*Sparus aurata*) [20] and striped bass (*Morone saxatilis*) [21]. Yet, anesthetics induce species-specific responses [18] that require further tests. On this subject, the objective of this study was to evaluate, for the first time, propofol as an alternative anesthetic to Nile tilapia submitted to a simulated transport for 6 h, as it has been described as suitable for the transportation of other species [26]. In general, propofol induced oscillations in sedation depth of the animals, while the control ones exhibited more erratic movements. By the end of the transportation, propofol decreased hematocrit (HCT) and lactate levels while increased glucose levels. Comparing with the naïve animals, MS-222 transported animals had a decreased HCT and lactate levels while cortisol levels increased. Cortisol levels also increased in the control animals and were still high 24 h later. Additionally, propofol induced a decrease in the gill ROS levels and GST activity by the end of the transportation when compared to the control group. Nevertheless, the quality of the fillet and the histopathology of the gills were unaltered in all tested groups.

Transported fish showed increased opercular movements immediately after immersion. These were normalized after a few seconds in the groups transported under sedation while respiratory movements of the control group were always higher, corroborating the light sedative anesthetic profile of these compounds for this species and transportation procedures [33,34]. Control fish had higher opercular movements than naïve animals, which may indicate stress induction during the simulated transportation procedure, except at 6 h, probably due to habituation to the surrounding conditions. In addition, in several time-points, control animals performed more erratic movements than animals treated with anesthetics, which advocates for the use of these compounds during transport, although the sedative effects can also contribute to this difference. In this context, animals transported under MS-222 showed no significant behavioral changes, while under propofol sedation, different responses were observed. Behavioral changes are of particular value for the welfare assessment of animals [60]. Propofol is a widely used anesthetic that targets the gamma-aminobutyric acid A (GABA_A_) receptors in the fish brain following rapid absorption through the gills into the arterial blood circulation [28,61,62]. It has recently been tested as an anesthetic for Nile tilapia [28] without reported side effects in this species [24]. Yet, in the current study, the fish transport simulated under propofol showed a decreased frequency of swimming and bottom swimming but spent more time swimming than the MS-222 group and were less inactive (frequency and duration of inactivity). These data contradict the observations of a lower opercular rate and less response to external stimuli, suggesting a variation in depth of sedation. This may be due to the propofol pharmacological properties or due to the fact that this compound is formulated as a lipidic oil-in-water emulsion [63], which may be not equally distributed throughout the water column, even if the solution is vigorously stirred prior to use [25]. Besides, there are reports of a paradoxical excitation with low doses of propofol characterized by a state of unexpected excitement of varying degrees, which may result from interactions between GABA_A_ receptor and intrinsic membrane potentials [64,65]. The meaning of this potential paradoxical excitation for the fish welfare is not clear, but it could actually be beneficial to the fish as, by maintaining a residual activity, fish can avoid physical damage from collision with objects or conspecifics during transportation. However, further pharmacodynamics and pharmacokinetics analysis are needed.

In addition to behavior, water quality is also an important factor to evaluate fish health during transportation procedures [6,14]. The transportation of Nile tilapia under MS-222 resulted in an increase in the water pH compared to the control group. Although ammonia levels were not measured representing a limitation of this study, increases in pH are associated with an increase in ammonia toxicity [66] with the accumulation of un-ionized ammonia, which is critical for transportation of fish [67]. Still, a recent study has shown a decrease of the total ammonia nitrogen (TAN) in a marine species (*Scophthalmus maximus*) transported for 6 h under the same MS-222 concentration [68]. In addition, the optimal water pH for the culture of this species varies between 5.5 and 9.0 [69,70] and the small variations observed might have no direct influence on the welfare of the animals. The hematocrit was decreased by the anesthetics after the end of the transportation, which could be an indicator of a decreased oxygen-binding capacity, associated with changes in pH [71,72]. However, no changes in hemoglobin were observed and the pH alterations were minor. Thus, the reduced hematocrit value observed was certainly not related to the pH changes. Nevertheless, a reduction of the hematocrit has previously been reported in Nile tilapia anaesthetized with propofol concomitant with a reduction in red blood cells and hemoglobin levels [28]. Still, similar and contradictory results have been reported in other fish species in which hemoglobin levels were both increased or decreased by propofol [73,74], warranting further clarification. MS-222 anesthesia has been described to increase hematocrit, but at higher concentrations than the one used in this study [33].

Notwithstanding, propofol induced an increase in glucose levels, while both propofol and MS-222 resulted in decreased lactate levels, which normalized, in both cases, after the recovery period. Furthermore, while control animals showed non-significant increased cortisol levels, animals from the MS-222 group had significantly higher cortisol values compared to those observed in the naïve group at 6 h. Cortisol and glucose are two of the most common stress indicators [75] and a stress response has previously been associated to anesthesia with MS-222 in this species [76]. The release of cortisol into the blood by interrenal cells of the head kidney following the activation of the hypothalamic-pituitary-interrenal (HPI) axis [77] triggers a wide range of secondary responses associated with energy homeostasis, growth, and osmoregulation [78]. In this regard, increased serum glucose is usually mediated by the action of cortisol in response to most stressors through the mobilization of energy reserves to cope with the energetic demands of the situation [79]. Although cortisol was not changed by propofol in the current study and despite glucose control and regulation varying greatly between organisms [80], the increase in glucose levels has been described as a common hallmark of propofol anesthesia in aquatic organisms [26,74,81,82]. Also, although not described for propofol, transport studies have shown a negative and delayed correlation between cortisol and glucose concentrations in blood plasma, with the cortisol peak being reached immediately after the stressful situation while glucose increased over time [83]. However, recent reports have also suggested a similar trend between cortisol and glucose plasmatic levels in the transport of Nile tilapia [84]. The increase of glucose at the end of transport could indicate stress caused by propofol. However, in addition to the lack of cortisol level alterations, and reduction of lactate values, no stress-related behaviors were observed. Overall, and although further clarification of these physiological changes is required, the data obtained support a non-stressed profile in the propofol group. MS-222 has been described to induce aversion in fish [18], and the less depth of sedation compared to the propofol group may induce some stress related to the confinement on these fish, thus showing increased cortisol levels at 6 h. However, the secondary and the tertiary stress response were not observed [85]: glucose levels were maintained at naïve levels, lactate levels decreased, and no relevant stress-related behaviors were identified. Thus, the fish exposed to MS-222 may not be aware of the cortisol increase, thus not suffering from stress. Nevertheless, the decreased lactate may be related to its anesthetic properties (i.e., the sodium-channel blockade effect [86]), which induces a lower muscle activity and, consequently, lower lactate plasmatic levels [87]. All groups recovered to levels similar to the ones from the naïve tilapia, except the cortisol of control groups that remained higher, showing, once more, the benefits of using anesthetics to minimize the stress resulting from transportation procedures.

Despite these changes, the organs’ functionality and the quality of the fillet were not affected by any of the treatments, supporting the use of both anesthetics for the transportation of this species. In accordance, a previous study has also shown no changes in tilapia (*Oreochromis aureus*) fillets following exposure for 30 min or euthanasia by carbon-dioxide anesthesia [16], further supporting the use of anesthesia for maintaining the quality of the final product. In the event of induced stress by the anesthetics supported by the changes in glucocorticoids, irregular oxidative responses would be detected in response to the activation of the HPI axis [88]. However, and similarly to what was previously reported under a comparable 6 h transportation of *Sparus aurata* [20], MS-222 did not interfere with the gills’ enzymatic antioxidant defenses, showing that the increase detected in cortisol may not have been problematic. Transport under propofol sedation induced a decrease in ROS levels and GST activity in the gills compared to control, while the muscle enzymatic defenses were irresponsive to both treatments. No differences were observed following the recovery period. Nevertheless, the decrease in ROS levels could be explained by the propofol antioxidant activity which has already been demonstrated both in vitro [89] and in vivo [90] due to its phenolic structure similar to that of alpha-tocopherol, a natural and potent antioxidant. Regarding the GST activity, a contrary study has found an increased gill GST activity following a 6 h sedation using a similar propofol concentration (0.8 mg L^−1^) in another aquatic animal [91], which was accompanied by an increase in SOD activity. GST plays an important role in the detoxification of xenobiotics thus protecting the cell from oxidative stress [92]. The decreased GST activity would be suggestive of a decline of the gills’ detoxification capacity which could be a consequence of the emergence of oxidative damage [93] or could be associated with a possible mechanism to compensate the ROS impact [94], but none of these were observed under propofol sedation. In this regard, the decreased GST activity might be justified by possible conformational changes of the protein following binding of propofol reducing GST conjugation ability. Although this has not been previously described in fish, propofol has been shown to inhibit this enzyme at high concentrations by this mechanism [95,96], further supporting this hypothesis. Despite this, oxidative stress parameters were not significantly affected by the anesthetics in the muscle, and minor alterations without physiological support were observed in the brain and liver of Nile tilapia transport simulated, in contrast to what has been previously reported for other species [20]. In general, there was no deleterious impact of sedation with these anesthetics on the antioxidant defenses of animals transported under this experimental setup. In fact, both anesthetics prevented the increase of the GST activity in the gills, the GSSG levels in the brain, and the GR activity in the liver observed in the control fish.

## 5. Conclusions

Overall, the few behavioral alterations and increase in glucose plasma levels observed at the end of the transport in fish treated with propofol could be associated with alterations of the sedative depth throughout the simulated transport, although the functionality of the organs and the welfare of the fish did not seem to be affected. However, propofol pharmacokinetics and solubility/stability need to be better studied for the live transport of cultured fish. In conclusion, the use of propofol sedation improved the transport conditions of Nile tilapia reducing stress-related responses and did not compromise its health, welfare, or fillet quality, providing a win-win situation to the animal, producers, and consumers, having potential implications for the aquaculture sector.

## Figures and Tables

**Figure 1 biology-10-01309-f001:**
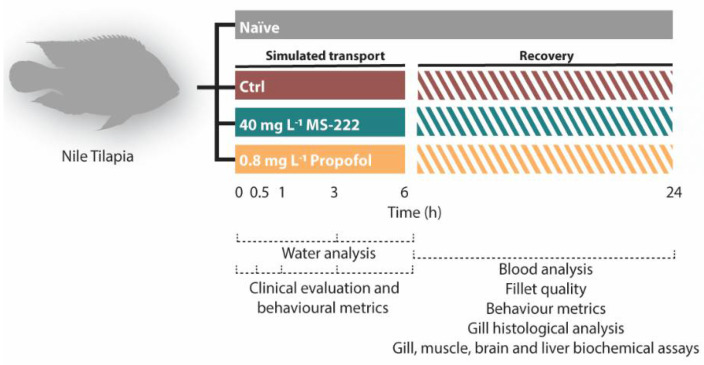
Timeline of the experimental design. Nile tilapia s were randomly distributed in naïve group (not transported) and into three transported simulated groups for 6 h: control (transported without sedation), MS-222 (transported with 40 mg L^−1^ MS-222) and propofol (transported with 0.8 mg L^−1^ propofol). During the transportation different endpoints were evaluated at 0, 0.5, 1, 3, and 6 h: clinical parameters, animal behavior, and the water chemical changes. At the end of the transportation and after a recovery period of 24 h, blood analysis, behavioral responses, fillet quality and histopathology of the gills were conducted. Different biochemical parameters were also evaluated in gills, muscle, liver, and brain.

**Figure 2 biology-10-01309-f002:**
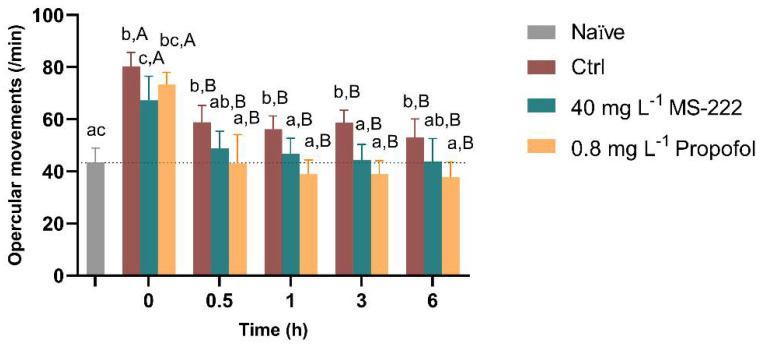
Opercular movement rate per minute of Nile tilapia during the simulated transport in the different experimental groups. Data are presented as mean ± SD of seven fish from each group. Different lowercase letters indicate differences between groups at a specific time-point, and capital letters represent statistical differences for the same group over time (*p* < 0.05).

**Figure 3 biology-10-01309-f003:**
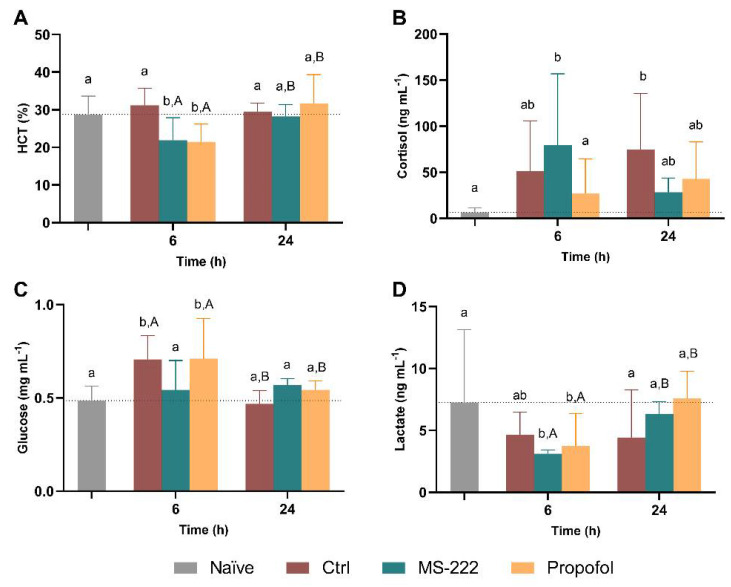
Hematocrit (HTC) (**A**), cortisol (**B**), glucose (**C**), and lactate (**D**) levels at the end of the simulated transport (6 h) and at the end of the recovery period (24 h). Data are presented as mean ± SD (**A**,**B**) or as median and interquartile ranges (**C**,**D**) from at least five independent replicates. Different lowercase letters indicate differences between groups at a specific time-point while capital letters represent statistical differences for the same group over time (*p* < 0.05).

**Figure 4 biology-10-01309-f004:**
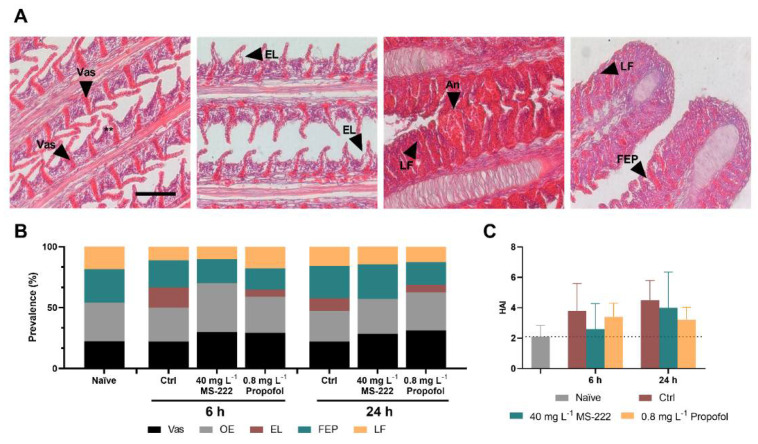
Representative histopathological changes observed in the gill epithelia of fish from the different groups (**A**). Gill filaments, lined by a stratified epithelium evidencing the presence of oedema (**), mainly in the deeper region of the epithelium and nearby lamellar vascular axis, showing some vasodilation (Vas) in its base (first image). Gill filaments with high severity level of oedema (**) that sometimes conducted to epithelial lifting (EL) (second image) and gill filaments showing filament epithelium proliferation (FEP) that, in some cases, led to lamellar fusion (LF) (third and fourth image) and higher severity of vasodilatation, that extended through the entire lamellar vascular axis; aneurisms (An) were sporadically observed (third figure). Haematoxylin-eosin staining; scale bar: 100 µm. Prevalence (%) of histopathological changes in gills over time in the different experimental groups (**B**) Vas, vasodilatation; OE, oedema; EL, epithelial lifting; FEP, filament epithelium proliferation; and LF, lamellar fusion. Histopathological alteration index (HAI) of Nile tilapia gills after transportation and after the recovery period (**C**). Data are presented as mean ± SD of seven fish. No statistical differences were observed between experimental groups at any of the analyzed time-points.

**Figure 5 biology-10-01309-f005:**
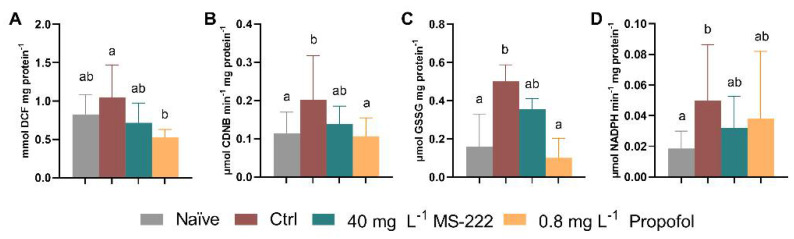
Changes in ROS levels (**A**), and GST activity (**B**) in the gills, GSSG levels (**C**) in the brain, GR activity (**D**) in the liver at the end of the simulated transport (6 h). Data are presented as median and interquartile ranges (**A**,**B**,**D**) or as mean ± SD from at least five independent replicates. (**C**) Different lowercase letters indicate differences between groups (*p* < 0.05).

**Table 1 biology-10-01309-t001:** Ethogram of the observed behaviors in adult Nile tilapia at 0, 0.5, 1, 3 and 6 h (0–6 h) of simulated transport and 24 h after the transport ends. The “x” signals the behaviors analyzed on each time-point.

Behavior	Description	0–6 h	24 h
Swimming (S)	Fish swims without touching the floor and using the fins (no contact between abdomen and floor, but fins may touch it).	x	x
Bottom swimming (BS)	Fish swims in contact with the tank floor using the fins.	x	x
Inactive (I)	Fish is in the tank floor without locomotion/movement in space.	x	x
Erratic movements (EM)	Sharp changes in direction or velocity and repeated rapid darting (fast acceleration in one direction with the use of caudal fin).	x	x
Air stone breathing (AB)	Fish is near/in contact with the air stone with its mouth pointed to this object.	x	
Turning (T)	Fish changes direction of movement.	x	
Rubbing (R)	Rubbing body sides on the sides of the tank (or on the surface of other objects).	x	
Crossings (C)	Number of times the fish crosses a virtual line. The tank was divided in 3 horizontal and 4 vertical zones by 2 and 3 imaginary lines, respectively.		x
Mirror stimulation (MS)	Head-butting (pushing head against the sides or bottom of the tank), biting these surfaces, or chasing own reflection in close contact with the tank sides or bottom.		x
Interaction with objects (IO)	Fish interacts by biting or actively touching the thermostat or thermometer or other objects inside the tank.		x

**Table 2 biology-10-01309-t002:** Number of fish responding to different external stimuli and water chemistry (pH, dissolved oxygen) from the three tested groups during the simulated transport procedure.

Timepoint (h)	Groups	Response to Stimulus			Water Chemistry	
		Visual ^1^	Touch ^1^	Pinch ^1^	pH	O_2_ (mg L^−1^)
0					7.33 ± 0.24 ^AB^	7.72 ± 0.03 ^AB^
0.5	Control	7/7 (100%) ^a^	7/7 (100%) ^a^	0/0(100%) ^a^		
	MS-222	5/7 (71%) ^ab^	4/7 (57%) ^a^	3/3(100%) ^a^		
	Propofol	2/7 (29%) ^b^	0/7 (0%) ^b^	4/7 (57%) ^b^		
1	Control	7/7 (100%) ^a^	7/7 (100%) ^a^	0/0(100%) ^a^		
	MS-222	6/7 (86%) ^ab^	5/7 (71%) ^a^	2/2(100%) ^a^		
	Propofol	2/7 (29%) ^b^	0/7 (0%) ^b^	4/7 (57%) ^b^		
3	Control	7/7 (100%) ^a^	7/7 (100%) ^a^	0/0(100%) ^a^	7.03 ± 0.22 ^a,A^	7.21 ± 0.52 ^A^
	MS-222	6/7 (86%) ^ab^	5/7 (71%) ^a^	2/2(100%) ^a^	7.62 ± 0.53 ^b,A^	7.54 ± 0.20 ^A^
	Propofol	2/7 (29%) ^b^	0/7 (0%) ^b^	4/7 (57%) ^b^	7.17 ± 0.25 ^ab,A^	7.10 ± 0.46 ^C^
6	Control	6/7 (86%) ^a^	7/7 (100%) ^a^	0/0(100%) ^a^	7.49 ± 0.27 ^a,B^	7.55 ± 0.30 ^B^
	MS-222	5/7 (71%) ^ab^	5/7 (71%) ^a^	2/2(100%) ^a^	7.81 ± 0.22 ^b,A^	7.62 ± 0.18 ^A^
	Propofol	1/7 (14%) ^b^	0/7 (0%) ^b^	5/7 (71%) ^b^	7.74 ± 0.11 ^ab,C^	7.69 ± 0.08 ^A^

^1^ Parameters quantified as the number of animals reacting to the stimulus per total of animals tested in each group and the respective percentage (%) of response; pinch was only tested in animals not responding to the touch stimulus. The binomial data presence/absence of response were analyzed statistically (n = 7). For the other parameters, data from the same independent replicates were expressed as mean ± SD for parametric data distribution. Statistical analysis was performed using repeated measures ANOVA followed by Tukey’s multiple-comparison test. Different lowercase letters indicate significant differences between groups while capital letters represent statistical differences between time within the same group (*p* < 0.05).

**Table 3 biology-10-01309-t003:** Frequency per min (#) and duration in seconds (time) of the behaviors analyzed during the simulated transport. Five min of recordings were analyzed in each time-point.

Timepoint (h)	Groups	S		BS		I		EM		AB		T	R
		#	Time	#	Time	#	Time	#	Time	#	Time		
0	Control	1.6 (0.2–4.7) h	60 (16–110)	0.6 (0.4–2.4) ^AB^	25 (5–74)	0.8 (0.6–1.8)	172 (85–232)	0.6 (0.2–1.8)	21 (15–30) ^a,A^	0.0 (0.0–0.2)	0 (0–60)	2.6 (0.8–7.1)	2.6 (1.4–3.6) ^A^
	MS-222	0.6 (0.4–2.4)	30 (4–70)	0.8 (0.0–2.2) ^AB^	53 (0–80) ^AB^	0.8 (0.2–2.0) ^AB^	213 (114–275)	0.4 (0.0–1.2)	16 (0–33) ^ab^	0.0 (0.0–0.2)	0 (0–13)	2.2 (1.0–5.7) ^A^	1.8 (1.0–3.6)
	Propofol	0.8 (0.2–3.9)	47 (3–113) ^A^	0.2 (0.2–2.7)	6 (2–113)	0.6 (0.2–2.2)	240 (46–277) ^A^	0.2 (0.2–2.3)	10 (8–19) ^b^	0.0 (0.0–0.6)	0 (0–6)	1.4 (0.8–5.0)	0.8 (0.2–5.2)
0.5	Control	2.4 (1.2–4.8) ^a^	86 (47–145) ^ab^	2.2 (0.4–2.6) ^a,A^	44 (4–86)	1.8 (0.8–2.2) ^a^	151 (95–210) ^a^	1.0 (0.0–1.8)	10 (0–42) ^AB^	0.0 (0.0–0.8)	0 (0–30)	2.0 (0.6–5.9) ^a^	2.0 (0.2–6.3) ^AB^
	MS-222	1.2 (0.2–2.4) ^ab^	24 (5–139) ^a^	1.6 (0.4–2.4) ^ab,A^	53 (29–101) ^A^	1.6 (0.8–2.0) ^ab,A^	205 (57–275) ^a^	0.4 (0.0–0.8)	4 (0–17)	0.0 (0.0–0.6)	0 (0–15)	1.4 (0.0–3.0) ^ab,AB^	1.2 (0.2–2.8)
	Propofol	0.2 (0.0–2.2) ^b^	134 (0–301) ^b,AB^	0.2 (0.0–1.9) ^b^	15 (0–237)	0.4 (0.0–1.7) ^b^	9 (0–141) ^b,AB^	0.0 (0.0–0.8)	0 (0–10)	0.0 (0.0–0.2)	0 (0–30)	0.4 (0.0–1.6) ^b^	0.0 (0.0–3.5)
1	Control	2.0 (0.4–4.6) ^a^	105 (7–188) ^ab^	1.2 (0.8–2.8) ^A^	48 (16–62)	1.4 (0.8–3.0) ^a^	124 (53–238) ^a^	0.6 (0.0–2.8) ^a^	13 (0–52) ^a,AB^	0.0 (0.0–1.0)	0 (0–112)	2.2 (1.0–4.2) ^a^	1.2 (0.6–4.4) ^AB^
	MS-222	0.0 (0.0–2.2) ^b^	0 (0–153) ^a^	0.0 (0.0–2.6) ^AB^	0 (0–68) ^AB^	0.4 (0.2–1.8) ^ab,AB^	300 (79–300) ^a^	0.0 (0.0–0.4) ^b^	0 (0–5) ^b^	0.0 (0.0–0.2)	0 (0–11)	0.0 (0.0–2.0) ^b,AB^	0.0 (0.0–1.8)
	Propofol	0.2 (0.2–1.0) ^b^	135 (6–301) ^b,AB^	0.2 (0.0–2.0)	62 (0–258)	0.4 (0.0–1.4) ^b^	4 (0–187) ^b,AB^	0.0 (0.0–0.6) ^b^	0 (0–11) ^b^	0.0 (0.0–0.2)	0 (0–6)	0.4 (0.0–1.6) ^b^	0.4 (0.0–3.6)
3	Control	2.0 (0.4–3.0)	68 (28–208) ^ab^	0.4 (0.0–1.6) ^B^	18 (0–40)	1.4 (1.2–2.4) ^a^	201 (51–259) ^a^	0.2 (0.0–1.2)	2 (0–22) ^B^	0.0 (0.0–0.4)	0 (0–220)	1.0 (0.2–2.8)	0.6 (0.0–2.4) ^B^
	MS-222	0.6 (0.0–2.6)	28 (0–95) ^a^	0.6 (0.0–2.4) ^AB^	28 (0–76) ^AB^	1.0 (0.2–2.0) ^ab,AB^	257 (119–301) ^a^	0.2 (0.0–0.8)	1 (0–10)	0.0 (0.0–0.4)	0 (0–33)	1.0 (0.0–1.6) ^AB^	0.6 (0.0–1.4)
	Propofol	0.8 (0.0–2.0)	239 (0–301) ^b,AB^	0.4 (0.0–1.2)	9 (0–62)	0.0 (0.0–0.4) ^b^	0 (0–300) ^b,AB^	0.0 (0.0–1.2)	0 (0–78)	0.0 (0.0–1.2)	0 (0–76)	0.4 (0.0–6.0)	0.4 (0.0–6.0)
6	Control	2.0 (0.8–3.5) ^a^	58 (40–81) ^ab^	1.0 (0.4–3.1) ^a,AB^	22 (5–82)	1.4 (1.0–2.9) ^a^	215 (93–238) ^a^	0.2 (0.0–2.3)	2 (0–39) ^B^	0.0 (0.0–0.6)	0 (0–123)	0.8 (0.2–4.2)	0.8 (0.0–2.7) ^AB^
	MS-222	0.2 (0.0–1.0) ^b^	16 (0–292) ^a^	0.0 (0.0–1.0) ^b,B^	0 (0–33) ^B^	0.2 (0.0–1.2) ^b,B^	267 (0–300) ^a^	0.0 (0.0–0.4)	0 (0–11)	0.0 (0.0–0.0)	0 (0–0)	0.0 (0.0–1.0) ^B^	0.4 (0.0–1.6)
	Propofol	0.4 (0.2–1.4) ^b^	298 (4–300) ^b,B^	0.0 (0.0–1.2) ^b^	0 (0–243)	0.0 (0.0–0.8) ^b^	0 (0–51) ^b,B^	0.2 (0.0–0.8)	2 (0–15)	0.0 (0.0–0.4)	0 (0–182)	2.0 (0.0–5.4)	0.4 (0.0–4.0)

Swimming (S), bottom swimming (BS), inactive (I), erratic movements (EM), Airstone breathing (AB), turning (T), and rubbing (R) obtained from seven independent replicates and expressed median and interquartile ranges. Statistical analyses were performed using the Kruskal–Wallis test followed by Dunn’s multiple-comparison post-hoc tests analysis while the effect of time was tested by Friedmann. Different lowercase letters indicate significant differences between groups while capital letters represent statistical differences between time within the same group (*p* < 0.05).

## Data Availability

All relevant data are within the manuscript and its Appendix A files and are available from the corresponding author on reasonable request.

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
