# Peer review of "MS-222 and Propofol Sedation during and after the Simulated Transport of Nile tilapia (Oreochromis niloticus)"

_biology, 2021, doi:10.3390/biology10121309_

Round 1

Reviewer 1 Report

Your work is adequate. Please amend few issues indicated in the attached file. 

Author Response

L38– insert “.”

Response: The dot was added to the sentence.

L200 – which equation?

Response: The equation for converting the absorbance of the solution to Hb concentration has been described previously by the cited reference. As such, the authors believe that there is no need to include it in the manuscript.

L201 - ?

Response: The authors apologize for the confounding word which has been changed to “clot”.

L223 – Measurements

Response: The word has been changed.

L248 - I don't get it. What do you mean? You cut off two whole gill arches of each fish? The fish survived such a severe operation? From each fish? Do you mean gill filaments instead?

Response: Regarding the questions raised by the reviewer, most fish have three or more gill arches on each side of the body which support the gill filaments. In this work, following euthanasia, two gill arches (one from each side of the body) were collected for the histopathological analysis as described. This was collected on death fish. In order to clarify this, the sentence was revised.

L294 - time of

Response: A space was added between the words.

L591 - Ammonium (NH4+} percentage is reduced at higher pH. Higher pH favors un-ionized ammonia (NH3) percentage. Do you mean ammonia here?

Response: The authors apologize for the confusing sentence. The reviewer is right to point out that higher pH favors the increase in the amount of un-ionized ammonia. The sentence was revised.

Reviewer 2 Report

Excellent MS. My very few comments are included in the text in the form of comments. To see them all, open the file in Acrobat Reader.

Author Response

L20 – Nile tilapia

Response: The fish common name was added.

L116 – add details: protein and fat level

Response: The authors appreciate the question raised by the reviewer. However, it was not the objective of this study to evaluate the effects of the diet on the live transportation of this species and the food was only used for animal maintenance reasons. In fact, transported animals were not fed during nor 24h before the experiment to avoid the interference of feeding. Still, as described in the cited reference, the control diet is composed of 35.8 % of protein and 8.3 % fat. These values were added to the manuscript as requested.

L117 – Did you measure DO and ammonia?

Response: Thank you for this question. Dissolved oxygen was measured throughout the experiment as described in L169. However, ammonia was not measured being this a limitation for this study as described in L636.

L131 – described more detaily

Response: The authors believe that the description of the naïve group is complete. In fact, the naive group was composed of animals which were maintained in the original tank and not being used in the simulated transport or for any of the other treatment; they were only euthanized and tissues collected for the analysis referred in the manuscript.

L157 – add n = ?

Response: The n was added.

L186 – n =?

Response: The n was added in the text in L173.

L347 – from each grup/repetition?

Response: The information was added to the figure caption.

L471 – title should be above the Table

Response: The authors apologize for this typo which is out of our control as this table was changed by MDPI in our submitted file. We will make our best to guarantee that the final version maintains the correct table and format.

Reviewer 3 Report

Review

Paper title: MS-222 and propofol sedation during and after the simulated transport of Nile tilapia (Oreochromis niloticus).

The authors conducted a complex laboratory study to reveal possible effects of MS-222 and propofol on Nile tilapia as a result of conditions simulating fish transport for 6 h and 24 h after exposure. Propofol has been demonstrated to increase swimming activity in the fish but led to decreased opercular movements. Increased levels were found for water pH and glucose levels, an opposite pattern was registered for the haematocrit and lactate. In general, there were no significant effects on the physiology and tissue conditions of fish. Moreover, the fillet quality was also not affected. The authors concluded that propofol may be considered an alternative to MS-222. These results may have important implications for aquaculture and transportation of Nile tilapia.

All these reasons explain the relevance of the paper by Luís Félix and co-authors submitted to "Biology".

General scores.

The data presented by the authors are original and significant. The study is correctly designed and the authors used appropriate methods. In general, the statistical analyses are performed with good technical standards. The authors conducted careful work that may attract the attention of a wide range of specialists focused on the fish physiology and aquaculture.

Recommendations.

References should be formatted according to Rules for Authors.

Specific comments.

L 34. Change “behaviour” to “activity”

L 36. Change “were increased” to “increased”

L 38. Change “levels” to “levels.”

L 39. Change “were lessened” to “lessened”

L 40. Change “were normalised” to “normalised”

L 58. Change “associated to” to “associated with”

L 68. Change “turbulence,” to “turbulence, and”

L 82. Change “report” to “research”

L 88. Change “alterations on” to “alterations in”

L 124. Change “distributed in” to “distributed into”

L 130. Change “a light sedation” to “light sedation”

L 136. Change “a short” to “short”

L 145. Change “an acute stress” to “acute stress”

L 151. Change “water chemical” to “water chemistry”

L 178. Change “a specific behaviour” to “specific behaviour”

L 181. Change “single blinded” to “single-blinded”

L 194. Change “calculated” to “was calculated”

L 205. Change “collected” to “was collected”

L 237. Change “After” to “Afterward”

L 246. Change “in direct” to “indirect”

L 275. Change “determined” to “was determined”

L 282. Change “were measured” to “was measured”

L 294. Change “timeof” to “time of”

L 321. Change “propofol exposed” to “propofol-exposed” here and throughout the text

L 323. Change “However, but” to “However,”

L 323-324. Change “compared with” to “compared to”

L 324. Change “naïve” to “naïve group”

L 324. Change “propofol treated” to “propofol-treated” here and throughout the text

L 325. Change “compared with” to “compared to”

Figures 2-4, Table 3. Change “Ctrl” to “Control”

L 359. Change “MS-222 sedated” to “MS-222-sedated” here and throughout the text

L 365. Change “MS-222 group” to “the MS-222 group”

L 373. Change “repeated measures” to “repeated-measures”

L 376. Change “behaviour” to “activity”

L 408. Change “control  group” to “the control  group”

L 432. Change “compared with” to “compared to”

L 463. Change “the Supplementary Table 2” to “Supplementary Table 2”

L 494. Change “high severity” to “a high severity”

L 516. Change “The OSI was” to “The OSI”

L 517. Change “from the 6 h” to “from 6 h”

L 547. Change “control” to “the control ones”

L 551. Change “compared” to “comparable”

L 552. Change “were  increased” to “increased”

L 590. Change “increases of” to “increases in”

L 591. Change “associated to” to “associated with”

L 592. Change “as shown” to “has shown”

L 596. Change “influence in” to “influence on”

L 611. Change “MS-222 group” to “the MS-222 group”

L 633. Change “propofol group” to “the propofol group”

L 650. Change “an induced stress” to “induced stress”

Author Response

References should be formatted according to Rules for Authors.

Response: The references were carefully revised.

Specific comments

Response: The authors are grateful for the time you have spent reviewing the article. All the changes were made according to the sent comments.